# Energy Efficiency and Thermal Comfort Analysis in a Higher Education Building in Brazil

**Elisabeti F. T. Barbosa [1,*], Lucila C. Labaki [1], Adriana P. A. S. Castro [1] and Felipe S. D. Lopes [2]**

[1] Graduate Program on Architecture, Technology and City, School of Civil Engineering, Architecture and Urban Design, University of Campinas (UNICAMP), Campinas 13083-889, SP, Brazil; lucila@fec.unicamp.br (L.C.L.); dripasc@gmail.com (A.P.A.S.C.)

[2] Departament of Science and Technology, Federal University of Amapá (UNIFAP), Macapá 49100-000, AP, Brazil; felipe.lopes@unifap.br

[*] Correspondence: elisabeti.barbosa@gmail.com

**Abstract:** Thermal comfort is extremely important in architecture, especially in environments with more people spending longer time on studies or intellectual activities. This research describes a case study designed to investigate the energy and thermal performance of university buildings as part of the ANEEL programme. Because of this importance and the need to save energy in Brazilian public buildings, ANEEL—the Brazilian Energy Electricity Regulatory Agency—launched a national programme focusing on energy efficiency in public universities in 2016. University offices and classrooms sustain high intellectual effort; thus, environmental comfort is critical for maintaining their users' physical and mental health. This study included a pre-diagnosis of the performance of the envelope, lighting, and air-conditioning systems and a survey about the quality of the environments from the user's point of view. The Prescriptive Method of the Brazilian Labelling Program (PBE) for Commercial, Service, and Public Buildings (RTQ-C) was used to assess the building performance. Statistical analysis was applied to correlate the quality and thermal preference of the users, with reference to the predicted mean vote and the predicted percentage of dissatisfied (PMV-PPD). The results showed a high rate of thermal discomfort in both study environments, even when using air conditioning.

**Keywords:** Brazilian labelling regulation; energy efficiency; thermal comfort; university buildings

## 1. Introduction

Energy consumption in buildings has grown significantly in recent decades, representing over 30% of the world's total consumption, mainly from fossil fuels [1]. Despite this high consumption, buildings often lack satisfactory environmental quality for users [2]. Thus, the architectural design must be guided by the principles of sustainable development, energy efficiency, and thermal comfort [3]. Large-scale national policies governing energy efficiency are already mandatory in several countries, even in residential buildings (e.g., Australia, Canada, France, Germany, and Spain) [4,5]. In Brazil, energy efficiency initiatives, the growing demand for supply, and the possible scarcity of natural resources have driven the development of environmental certification and labelling programmes [6].

Building energy standards and labelling are key public policies that help accelerate the shift from conventional buildings towards sustainable low- to ultra-low-energy buildings. In the United States, the implementation of the Energy Star certification is recognised by the housing market and has produced a 25% total energy reduction in buildings [7]. The Energy Performance of Buildings Directive (EPBD) is a mandatory certification for European Union (EU) member states. China has been committed to researching and promoting energy-conservation standards, including assessment standards for green buildings [7].

The concern about fostering a sustainable campus has existed for over a decade. For example, a handy toolkit to connect theory and practice is found on the UNEP website [8]

and was developed based on the international collaboration of several universities. The preface to this document highlights that "universities, as the pinnacle of formal, organised education, have a particular responsibility both to help define and also to become exemplars of environmental best practice". Furthermore, many publications highlight the campus as a living laboratory where students and other users' involvement can transform the environment. In addition, a more general concept was suggested by Martinez-Acosta [9], from the perspective of universities as laboratories of sustainable cities.

In a more general trend, these developments on campus sustainability indicate that academia is now experiencing a growing awareness of the implications of modern, post-industrial civilisation for higher education (Aver et al. [10]). A significantly different knowledge base and set of abilities are needed to deal with the intensity and complexity of 21st-century living, particularly the sociopsychological characteristics of young people preparing to enter the workforce. Universities are now also expected to promote the skills and drive and to instil in young people the ideals of sustainable development and social responsibility.

Dawodu et al. [11] reviewed over 1000 articles on campus sustainability and assessment tools to identify the gaps, trends, and focus areas of campus sustainability that can be assessed by existing Campus Sustainability Assessment Tools (CSATs). They found 15 dimensions that govern the design of sustainable campuses, and the predominant dimensions were environmental, educational, and governance. The authors emphasised the importance of the building dimension in campus sustainability as the physical driver of sustainable education. Other building dimension issues include green-friendly considerations such as the green office, green lab, and others. These green spaces are renowned for fostering learning environments that are intelligent, individualised, and adaptive, which can lessen the cognitive burden on students. According to Dawodu et al. [11], another advantage is the significant reduction in energy consumption that may be gained from energy-saving building strategies. Additionally, campuses serve as symbolic environments for culture and society and as physical workplaces for employers.

Within this context, the University of Campinas (UNICAMP) and the CPFL Energy Company partnered to establish a management and energy efficiency model that can be replicated in other higher education institutions in Brazil and Latin America [12]. The Sustainable Campus Project included a labelling subproject based on the RTQ-C regulation to evaluate the energy efficiency of the UNICAMP buildings.

This research investigated users' energy behaviour and thermal comfort in a Brazilian public higher education building complex following the country's Labelling Program in Buildings. The selected buildings in this research are located on the main campus of the University of Campinas, in Campinas City, state of Sao Paulo, Brazil. An evaluation of thermal comfort analysis included on-site measurements of indoor thermal conditions and a survey of the building users. Questionnaires were distributed to users in representative environments, according to the criteria of the Performance Standard ASHRAE 55 [13]. In addition to data collection for the qualitative analysis of environments, observations and information notes related to constructive technologies (windows and doors) and air conditioning were also used.

## 2. Literature Review

The literature review section is organised into three topics that supported the research development: (a) the Brazilian energy labelling regulation, (b) indoor thermal comfort, focusing on university buildings, and (c) occupant behaviour related to building use.

### 2.1. Energy Labelling for Buildings in Brazil

Following global concern prompted by the 1970s energy crisis, the Brazilian Electricity Conservation Program (PROCEL) was launched in 1985 to encourage energy savings in different areas. In Brazil, buildings account for 51.2% of the electricity consumption, half of it consumed by the commercial and public sectors [14].

PROCEL and the Institute of Metrology, Standardization, and Industrial Quality (INMETRO), through the Labelling Program in Buildings (PBE Edifica), launched the Regulation for Energy Efficiency Labelling of Commercial, Service and Public Buildings (RTQ-C) in 2009 and of Residential Buildings (RTQ-R) in 2010 [7,9,15]. PROCEL ushered in a new set of dynamics guiding the quest for architectural solutions, supplementary lighting, and air-conditioning projects that would upgrade buildings' thermal and energy performances [16].

The existing labelling programme is intended to become mandatory for all types of buildings in the coming years [17]. Currently, in Brazil, labelling is compulsory only for federal public buildings over 500 $m^2$ to achieve Level A in the RTQ-C [12,18]. Regarding public universities, fewer than 10% have invested in building performance programmes and policies, with incentives from the Brazilian Electricity Regulatory Agency (ANEEL). When evaluating the energy performance of a public higher education building in Porto Alegre, southern Brazil, a study by Tomazi et al. [19] found the need to improve solar control on windows.

According to Pereira et al. [20], window components can increase the heat losses in winter and the solar gains in summer. The impact of windows is proportional to the area they occupy in façades [20]; windows are among the first building elements that demand attention to the implementation of energy retrofitting measures. The energy use is related to transparent surfaces (e.g., windows) affecting building climate control (Tong et al. [21]). According to Grynning et al. [22], transparent windows are responsible for about 37% and 40% of the total solar heat gain and heat loss through the building envelope, respectively. Conventional building envelopes, typically with numerous transparent elements such as static windows, cannot adapt to variable climatic conditions and changing user preferences (Tong et al. [21]). Excess energy demand for daylighting and conditioning is required in buildings to maintain the desired user comfort, thus resulting in a significant energy demand. Nevertheless, Moraes and Fonseca [23] highlighted the potential of automation to reduce energy use and promote energy efficiency in public buildings. According to Homod et al. [24], passive comfort in the building exploits the advantages of the natural climate to improve the conditions for occupants. A combination of efficient heating and cooling system factors achieves this. Nowadays, reducing the building's energy consumption is highly recommended.

According to Trgala [25], the thermal characteristics of buildings are currently mostly calculated by simplified mathematical models of the building envelope behaviour and by considering the thermal conductivity of individual building construction materials.

According to Spinelli et al. [26], based on the RTQ-C, it is possible to evaluate and classify the efficiency levels of an educational building by applying natural and innovative materials to the thermal insulation of wraps. Based on two methods, the prescriptive method and the simulation method, the Regulation for Energy Efficiency Labelling of Commercial Buildings (RTQ-C), launched by PROCEL in 2009, classifies buildings at five levels: from "A" (most efficient) to "E" (least efficient) [27]. The prescriptive method uses the results of simulations of building prototypes for each Brazilian bioclimatic zone, where multi-linear regression calculates the energy performance of the building envelope, considering building geometry, window-to-wall ratio, glazing solar factor, and shading devices. It also considers the building's lighting power density and air-conditioning system according to the equipment efficiency level.

The simulation method compares the proposed building with a reference model. The RTQ-C requirements are the normative reference for modelling to achieve the intended efficiency level. The same programme and weather data are input for simulations of the proposed and reference buildings. The proposed building energy consumption should be equal to or lower than that of the reference building to comply with the code specifications of the efficiency level.

Some studies assessed the methods used by RTQ-C and indicated the limitations of the prescriptive method regarding the building shape, envelope thermal properties, and window-to-wall ratio [15,28]. The regulation is under revision, and a new method is being developed using an artificial neural network to evaluate energy efficiency in commercial buildings [29]. Optimisation methods and machine learning are promising techniques that can be incorporated into energy policies. Furthermore, sustainable design extends beyond energy efficiency. In educational and office buildings, thermal comfort analysis is particularly interesting, as people spend around one-third of their time in this environment [30,31]. During this considerable amount of time spent indoors, occupants interact with the different systems of air conditioning, windows, blinds/shades, lighting fixtures, and others to meet their needs and preferences regarding indoor environmental quality (IEQ) [5,32]. The past decade was marked by an extensive research interest in the subject due to the impact of comfort on well-being, health, and productivity [32,33].

*2.2. Indoor Thermal Comfort*

Over the last decades, some standards, such as ASHRAE 55 and ISO 7730 [34], have been developed as technical references for maintaining thermal comfort inside environments. Thermal comfort field assessments are placed in two main categories to estimate the occupant's thermal comfort: (a) group 1—methods based mostly on the measurement of physical data, called the Objective Survey by Zomorodian et al. [35]; and (b) group 2—methods based on the occupant's perception of thermal comfort, collected with a questionnaire called the Subjective Survey by Zomorodian et al. [35], and also adaptative thermal comfort (ATC). The PMV (predicted mean value) is the primary approach of group 1, including four environmental factors (usually the radiant temperature, air temperature, air velocity, and relative humidity) and two human parameters (clothing level and metabolic rate).

The primary tool of group 2 studies is the questionnaire, which covers many thermal comfort issues. Early assessments asked only about preferences and feelings related to thermal comfort; however, at the moment, questions about indoor air dryness and air velocity are also being asked. Most field studies included descriptive scales such as the seven-point ASHRAE or Bedford scales for rating thermal sensation, the three-point McIntyre scale for thermal preference, and clothing and activity checklists (Mishra and Ramgopal [36]).

Both groups of models (objective/rational and subjective/adaptative) have been extensively used to evaluate thermal comfort. For applications in educational buildings, according to the review undertaken by Zomorodian et al. [35], neither methodology can individually predict students' thermal comfort accurately since, in contrast to office workers, students' adaptive mechanisms in educational buildings significantly influence their thermal preferences based on changes in activity and clothing level.

According to previous studies by Azadeh and Nicol [37], the evaluations of many investigated indoor spaces do not describe the thermal comfort requirements of occupants in a study space such as a classroom. In other words, the performances of classrooms and educational buildings did not reach a good satisfaction level according to the occupants' expectations, as highlighted in many studies mentioned below.

Several authors and studies reported the crucial importance of thermal comfort for human health, both physical and well-being. From a large-scope state-of-the-art review of occupant comfort in buildings, Faraji et al. [25,38] concluded that thermal comfort is more important than other aspects of occupant comfort, and evaluations of the interrelationship between occupant comfort and physiological, environmental, and psychological parameters have used regression-based approaches on most prediction models [39].

Zomorodian et al. [35] found that the lack of awareness and education is the most critical barrier to progress in energy conservation because of its significant influence on the attitude and behaviour of energy consumers. They presented an overview of thermal comfort field surveys in educational buildings over the last five decades, up to 2015. Most

of the studies concluded that students' thermal preferences were outside the comfort range provided by the standards, and reviewed studies showed that students prefer cooler environments and are more sensitive to warm conditions.

Guevara et al. [40] undertook a thermal comfort study in tropical (Ecuador) university classrooms. They concluded that the PMV model underestimated the occupants' satisfaction with the indoor conditions for air-conditioned environments. Several free-running and air-conditioned classrooms were part of the sample in this study. High comfort levels were reported in free-running classrooms in Quito, regardless of the lower air temperature. In contrast, students' classroom preferences in hot–humid climates are mechanically conditioned towards colder environments.

Zhang et al. [7] investigated the current status of the thermal environment and thermal comfort in the classrooms of Northeastern University (China) during the heating season. The indoor thermal environment was analysed using field measurements, a subjective questionnaire, regression statistics, and the entropy weight method. Increasing indoor relative humidity can effectively improve the overall thermal comfort of subjects. Furthermore, the temperature preference of women was higher than that of men. Additionally, they found the same conclusion as Guevara et al. [40] once the actual thermal neutral temperature was 2.5 °C lower than the value of the predicted mean vote. Faraji et al. [38] mentioned that other parameters, such as gender, outdoor temperature, and age, could play important roles in thermal comfort evaluation, although they are not included in several thermal comfort models.

Selected studies identified the differences in thermal perception depending on the building environment. According to Elnaklah [33,41], students' thermal perceptions inside educational buildings are affected by psychological adaptation, the duration of lectures, the student's gender, educational level, building operation mode, and climatic zone.

Satisfactory environmental quality, including thermal comfort, can affect the work performance and well-being of students and teachers since it helps improve productivity, learning ability, the user's systematic thinking, and building energy savings. On the other hand, thermal discomfort and poor air quality encourage demotivation and dissatisfaction and distract students and teachers. These aspects must be considered in the design and use of educational buildings since people spend much of their daily lives indoors [42–44].

### 2.3. Occupant Behaviour

Occupant behaviour is a significant factor contributing to the uncertainty in building energy usage prediction and causing inaccuracies in consumption forecasts. Consumption depends on occupants' interactions with the building or rather on their energy-related lifestyles. For several researchers [44–46], the most influential key variable in energy consumption is the variation in behavioural patterns related to the interaction with the space heating/cooling setpoints, equipment energy use, lighting, ventilation rates, and adjustment of the window openings, blinds, and shading devices.

Rashidzadeh and Matin [47] pointed out that the advancements in control technologies and material science enable the use of smart windows in high-performing façades to improve the building's energy performance and users' comfort. The authors recommended window choice based on its properties (reflectance, thermal transmittance rate, solar transmittance ratio, and solar heat gain coefficient) and the specific climatic zones of buildings. Hu et al. [48] highlighted the need for research to properly integrate occupant behaviour (OB) and building energy policies (BEP). They identified key questions and challenges related to technical standards and regulations, building information policies, energy incentives, and policy evaluations. They emphasised that uncertainties in services and the diversity of OB cause uncertainties in BEP design, implementation, and final effects.

Mamani et al. [43] identified the variables that influence the thermal comfort of a building, evidencing that the most used variables are those used by the predicted mean vote (PMV) index [49]. However, OB research has focused on new variables, seeking answers to individual differences in thermal perception, which can ultimately affect building energy

consumption, and aiming to improve people's well-being and quality of life in a sustainable framework [50]. Holanda [51] conducted a monitoring study, combined with computer simulation and parametric analysis, to investigate the impact of air-conditioning setpoint temperatures on thermal comfort for schools in hot and humid climates in Brazil. The research confirmed that design parameters (orientation, façade absorptance, and opening shadings), combined with HVAC capacity, can reduce energy consumption by up to 34%.

Hoyt et al. [28] examined the benefits of widening HVAC setpoints in seven office buildings of ASHRAE climate zones. Increasing the cooling setpoint from 22 °C to 25 °C saved an average of 29% cooling energy without reducing thermal satisfaction levels. Further widening temperature bands achieved with fans or personal controls can result in 32% to 73% HVAC savings, depending on the climate. All OB variables combined with thermal comfort and energy efficiency analysis can be incorporated into labelling programmes for a more comprehensive approach in a sustainable framework.

## 3. Materials and Methods

Given the research objective, this work presents the case study of a building energy performance analysis using the RTQ-C prescriptive methodology to diagnose the performance of the envelope and the air-conditioning equipment in a Brazilian subtropical climate context. The RTQ-C (technical requirements of the quality for the level of energetic efficiency of commercial, service, and public buildings) is the Brazilian programme of the INMETRO (Instituto Nacional de Metrologia, Normalização e Qualidade Industrial) PROCEL "Edifica" (national programme for energy efficiency in buildings).

Thermal sensation and preference were investigated, and the acceptable level of building environmental comfort and percentage of users' acceptance were determined. The predicted mean vote (PMV) determined by Fanger's method was used for the thermal comfort research. A short description of the RTQ-C methodology is presented below. A more detailed description can be found in Spinelli et al. [26].

### 3.1. RTQ-C Prescriptive Method

The RTQ-C regulation classifies the energy efficiency of a building using three main categories of assessment: lighting system, expressed by the internal lighting power density (LPD), air-conditioning system (AC), and building envelope (ENV). The three main categories can be evaluated using prescriptive or simulation methods (Wong and Kruger) [52]. Every requirement must be evaluated separately according to the specific assessment procedures. The final classification considers a 30% LPD, 40% AC, and 30% ENV weighting distribution [15]. The total score (PT) defines the final classification of the building (A to E), as depicted in Table 1.

**Table 1.** Final label levels of a building according to RTQ-C.

| Total Score (PT) | Final Label |
|---|---|
| 5 to PT $\geq$ 4.5 | A |
| 4.5 > PT $\geq$ 3.5 | B |
| 3.5 > PT $\geq$ 2.5 | C |
| 2.5 > PT $\geq$ 1.5 | D |
| PT < 1.5 | E |

Source: RTQ-C.

Equation (1) calculates the total score (PT) composed of a ratio between weights (established by end uses) for each system and the numerical equivalent of its partial efficiency level.

The prescriptive method employs a simplified framework to assess the building's energy efficiency level based on actual buildings from a sample of different Brazilian cities [28]. According to the Brazilian bioclimatic zone where a building is placed, the envelope analysis considers the geometric elements of the building (e.g., form factors, shading components, and façade orientations) and the material properties for wall and roof designs.

$$
\begin{aligned}
PT = \quad & 0.30 \times \left\{ \left( EqNumEnv \times \tfrac{AC}{AU} \right) + \left( \tfrac{APT}{AU} \times 5 + \tfrac{ANC}{AU} \times EqNumV \right) \right\} + 0.30 \times (EqNumDPI) \\
& + 0.40 \times \left\{ \left( EqNumCA \times \tfrac{AC}{AU} \right) + \left( \tfrac{APT}{AU} \times 5 + \tfrac{ANC}{AU} \times EqNumV \right) \right\} + b
\end{aligned}
\tag{1}
$$

Here, the following applies:

- *PT* is the total score;
- *EqNumEnv* is the numerical equivalent of the envelope;
- *EqNumDPI* is the numerical equivalent of the lighting system;
- *EqNumCA* is the numerical equivalent of the air-conditioning system;
- *EqNumV* is the numerical equivalent of unconditioned zones;
- *APT* is the floor area of non-conditioned transitory permanence zones;
- *ANC* is the floor area of long-term non-conditioned zones;
- *AC* is the floor area of conditioned zones;
- *AU* is the usable area;
- *b* is the score obtained by the bonuses, which can vary from 0 to 1.

The RTQ-C method is also available online at "https://labeee.ufsc.br/sites/default/files/webprescritivo/index.html (accessed on 10 October 2023)"whose objective is to automate the building assessment procedures.

The tool is self-explanatory and can be input with details of the project design and construction materials. This tool may potentially help improve the energy performance of buildings. The first screen of the WebPrescriptive Tool is shown in Figure 1.

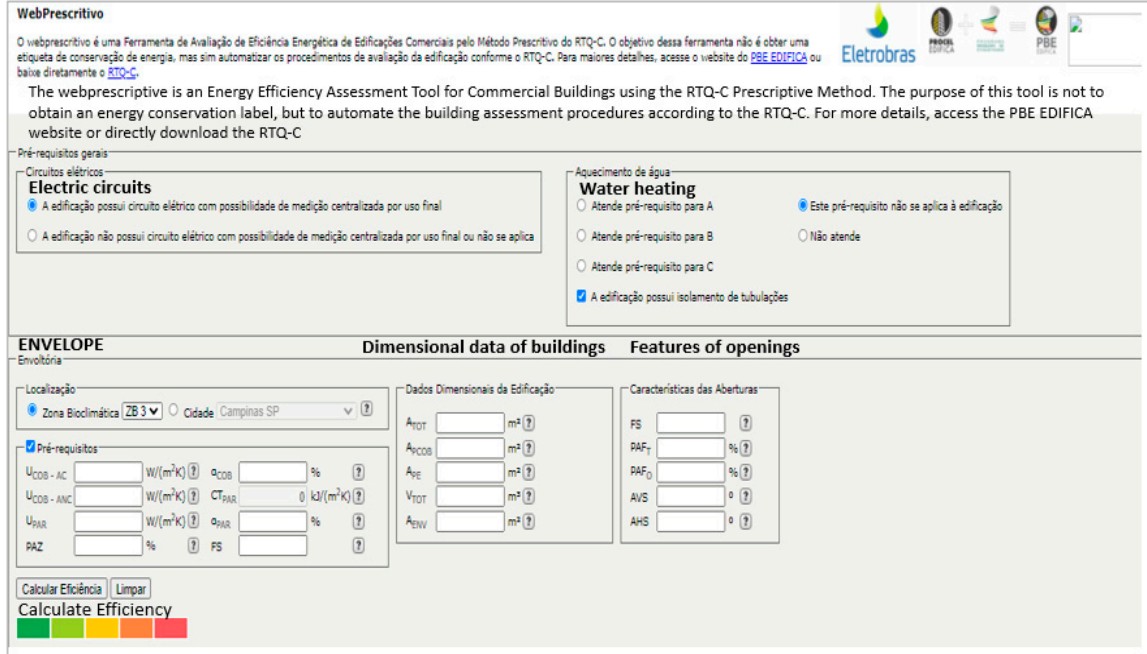

**Figure 1.** WebPrescriptive interface (RTQ-C prescriptive method tool).

### 3.2. Building Description (Geometry, Envelope, Lighting, and Air Conditioning)

The selected buildings in this research are located on the main campus of the State University of Campinas, District of Barão Geraldo. The School of Mechanical Engineering

(FEM, in Portuguese) complex is composed of 10 (ten) blocks divided into 2 distinct building sets (Figure 2). The research work took place in Building 2, which includes blocks H, I, J, and K. All blocks have three floors, in addition to transition blocks. The buildings' internal divisions are quite varied, following different rhythms and divided into spaces for workshops, laboratories, classrooms, teachers' rooms, meeting rooms, administrative spaces, library, auditorium, and social integration areas.

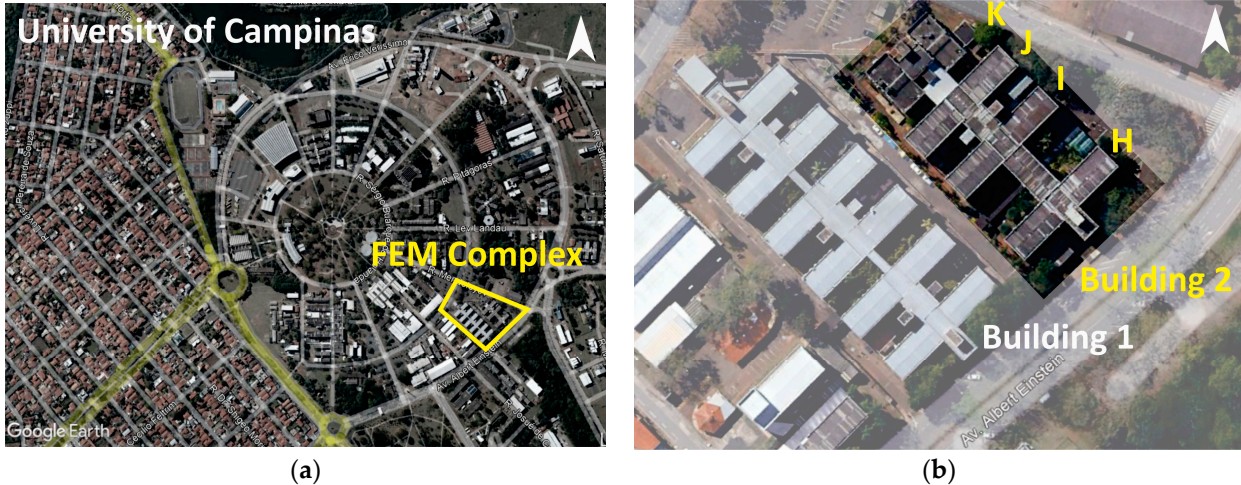

(**a**)  (**b**)

**Figure 2.** Building location: (**a**) the University of Campinas; (**b**) the School of Mechanical Engineering (FEM) complex, with emphasis on the analysed blocks.

Altogether, the blocks of Building 2 comprise 6877 m$^2$ of built area, and Table 2 describes the envelope materials. The walls are made of concrete blocks, partially painted and partially covered with ceramic tiles. The roofs are made of concrete slabs with fibre cement tiles. The windows are made of single glass with an average window-to-wall ratio of 38.1%.

**Table 2.** Envelope material properties of the buildings.

| Element | Properties |
|---|---|
| External walls | Exterior plaster (25 mm)/concrete block (90 mm)/interior plaster (25 mm) (U[1] = 2.92 W/m$^2$K \| $\alpha$[2] = 0.52) |
| Floors | Concrete slab (30 mm)/mortar (25 mm)/tile (10 mm) |
| Internal ceilings | Plaster (15 mm)/concrete slab (70 mm)/air gap/gypsum liner (10 mm) |
| Roof | Concrete slab (100 mm)/air gap/fibre cement tile (8 mm) (U[1] = 2.06 W/m$^2$K \| $\alpha$[2] = 0.52) |
| Windows | Simple glazing (U[1] = 5.80 W/m$^2$K \| SHGC[3] = 0.80) |

[1] thermal transmittance; [2] solar absorptance; [3] solar heat gain coefficient.

### 3.3. Climate Region

The FEM building is in Campinas, São Paulo, Brazil, at 22°54′ south latitude, 47°03′ west longitude, and 685 m altitude. It is in a dry-winter humid subtropical climate region (Cwa), according to the Köppen–Geiger classification [53]. The city's average annual temperature is 22.4 °C, with an average annual rainfall of 1424.5 mm. The rainy season is from October to March, with August being the driest month. The average relative humidity in Campinas ranges from 67% in January to 36% in August [13] (Figure 3).

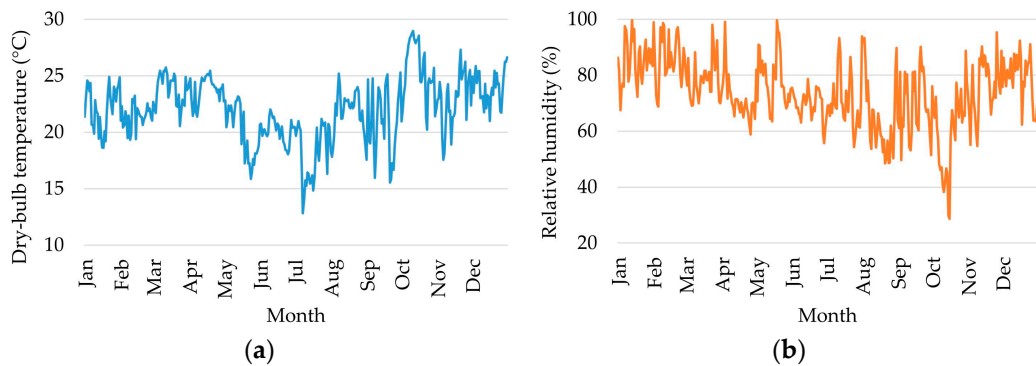

**Figure 3.** Climate characteristics of Campinas: (**a**) monthly average temperature; (**b**) monthly average relative humidity.

*3.4. Thermal Comfort Analysis*

An evaluation for the thermal comfort analysis included on-site measurements of indoor thermal conditions and a survey of the building users [42]. Nine conditioned rooms were selected for analysis (two classrooms and seven offices), as shown in Figure 4. The rooms were selected based on availability, according to the building manager. Both measurements and surveys occurred between 13 November and 14 December 2018, during the transition seasons between spring and summer in Campinas.

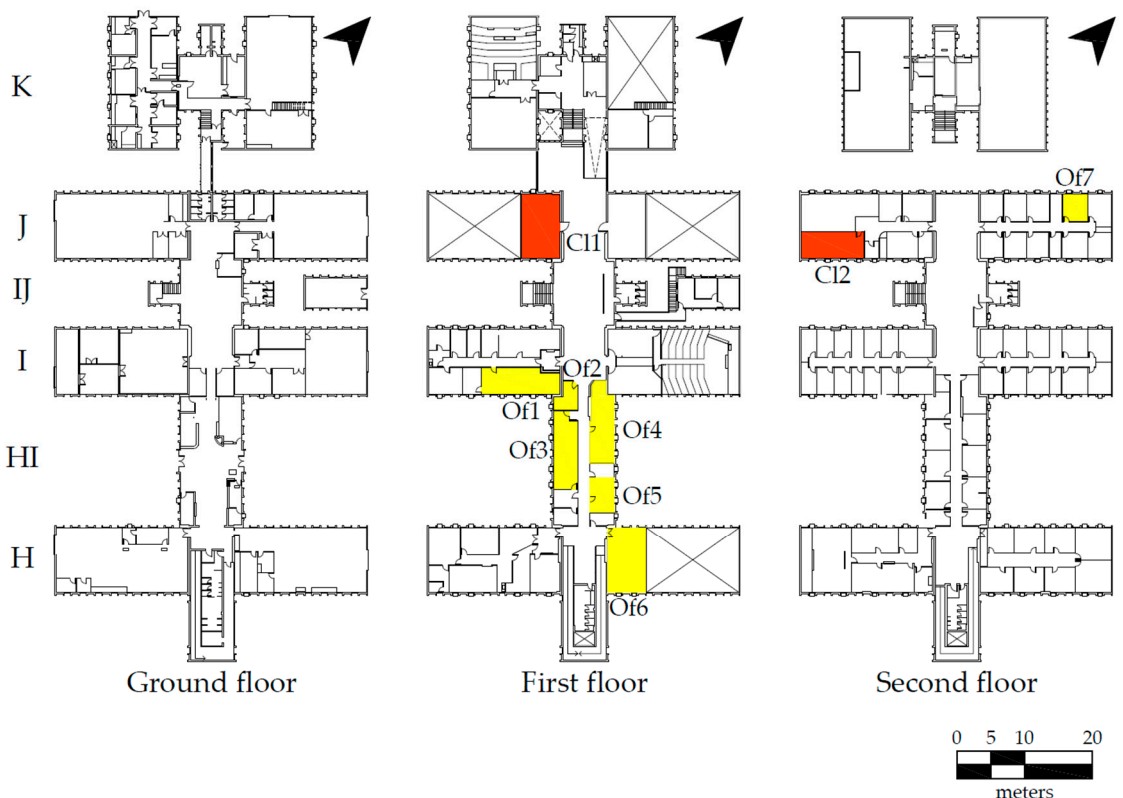

**Figure 4.** Floor plan of the FEM building, highlighting the selected classrooms (Cl1 and Cl2) and offices (Of1 through Of7).

3.4.1. On-Site Measurements

The environmental parameters obtained during the monitoring period were the relative humidity, air and globe thermometer temperatures, and air velocity, measured using the microclimate data logger Testo brand digital thermo-hygrometer, model 175-H1; Testo brand digital thermometer, model 175-T2; and Testo brand digital anemometer, model

405V, manufacturer (TESTO), Campinas, SP, Brazil. ASHRAE (2017) recommends that air temperature and average air velocity Va should be measured at 0.1, 0.6, and 1.1 m for seated occupants. However, due to the spatial configuration of the rooms, especially the offices, and as a way of ensuring uniform data collection, it was necessary to use the equipment at the greatest height indicated by the standard. The equipment used was fixed on a tripod 1.10 cm above the ground to simulate the sitting position of the users and far away from the external walls but near the user's seat. The instruments were shielded from direct solar radiation, cleaned, and calibrated.

The environmental and personal information collected in the buildings was input to evaluate the predicted mean vote (PMV) and predicted percentage of dissatisfied (PPD) calculated by using the computer program Comfort 2.03 [54], with the version updated in 2018.

### 3.4.2. Questionnaires

According to ASHRAE 55 [55], thermal comfort is "that condition of mind that expresses satisfaction with the thermal environment and is assessed by subjective evaluation". Due to its subjectivity, the environmental conditions required for comfort differ for everyone.

The questionnaires applied to selected rooms with different locations, dimensions, geographic orientations, floors, window ratios, air-conditioning equipment, and users. The characteristics of the survey are shown in Table 3. The window–wall ratio in some rooms reached 77.75 (%). This is a commercial building; however, the HVAC equipment (split or windows) is regularly adapted. Most buildings are around 30 years old.

**Table 3.** Characteristics of the selected rooms.

| Room | Area (m$^2$) | Floor Level | Orientation | Window-to-Wall Ratio (%) | HVAC | Total of Users in the Environment | Number of Responded Questionnaires |
|---|---|---|---|---|---|---|---|
| CL1 | 54.4 | 1st | NW-SE | 22.73 | S | 52 | 52 |
| CL2 | 37.1 | 2nd | SE | 22.73 | S | 45 | 45 |
| OF1 | 43.3 | 1st | SE | 22.73 | W | 3 | 10 |
| OF2 | 12.1 | 1st | SW | 77.5 | W | 4 | 16 |
| OF3 | 39.8 | 1st | SW | 77.75 | W | 5 | 21 |
| OF4 | 39.6 | 1st | NE | 77.75 | W | 3 | 12 |
| OF5 | 18.4 | 1st | NE | 77.75 | W | 3 | 12 |
| OF6 | 54.3 | 1st | SE | 17.2 | W | 4 | 16 |
| OF7 | 15.4 | 2nd | NW | 17.2 | W | 4 | 16 |

Note 1: S refers to split air-conditioner equipment, and W refers to window equipment. Note 2: each user would have to respond to 4 questionnaires per day, but some users did not respond completely.

### 3.4.3. Survey Study

The survey included a field study involving a set of five buildings that, inside, featured nine selected representative rooms for the thermal comfort study.

These buildings were from the University in Campinas—UNICAMP. The selected rooms used for research were teaching and administrative offices. The research occurred during the transition seasons between spring and summer, a warm time in the city. In Brazil, mechanically heated buildings with access to windows for natural ventilation are typical. Brager and Baker [56] referred to this combination of natural ventilation, window operation, and mechanical systems to provide cooling and obtain thermal comfort as a "Mixed mode". The surveyed buildings function in this way.

To collect environmental data, the questionnaire surveys were distributed simultaneously at pre-established times and through observation, and they aimed to describe the leading construction technologies (windows and doors) and equipment (air conditioning).

The research objective at the university was to describe the real (as built) way of using environments and air-conditioning technologies.

The questionnaire encompassed three steps:

Step one: general building information (local, room, time, kind of users, or other).

Step two: student information, such as age, gender, height, and the insulation based on ASHRAE 55 (2017) [55], clothing vestment (clo), and activity (met).

Step three: evaluation of the thermal environment, including the following questions:

List of questions about thermal preference vote (PPD) and thermal sensations vote (PMV) in the representative rooms.

Questions:

Q1: What is your thermal sensation at this moment? This references the temperature from −3 cold to +3 hot. It is symmetrical by zero, which is considered neutral, where a person feels comfortable.

Q2: How would you prefer the temperature to be in this environment right now? The reference intensity ranges from no change to much warmer or much cooler, as shown in Figure 5.

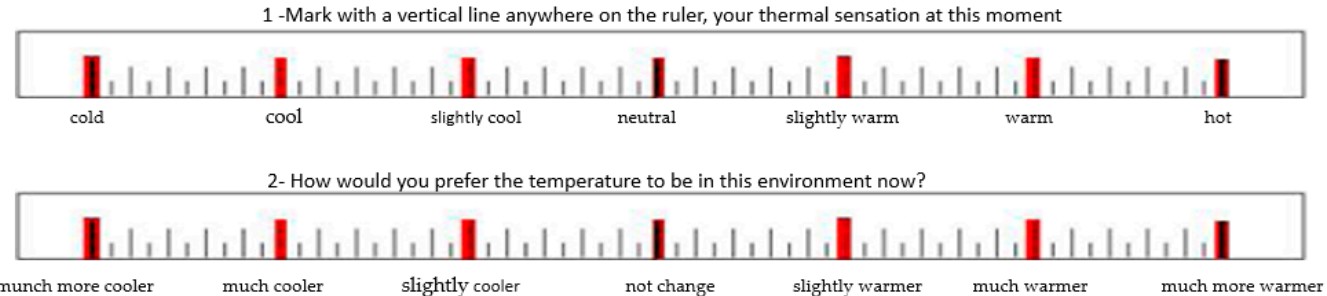

**Figure 5.** The questions submitted to the users.

Questionnaires were distributed to users in representative environments, following the criteria of the Performance Standard (ASHRAE 55, 2017 [55], page 16), or where they spend their time. In addition to data collection for the qualitative analysis of environments, observations and information notes related to constructive technologies (windows and doors) and air conditioning were also used.

To understand the relationship between the environmental and personal variables, a descriptive analysis of the collected data and a statistical inference method were carried out, seeking relationships between thermal performance and the types of environmental use.

The statistical sample was defined using Cochran's equation (1997) based on the work of Luca and Silva [57] according to Expression (2):

$$n = \frac{N.p.q.\left(Z_{\alpha/2}\right)^2}{(N-1).d^2 + p.q.\left(Z_{\alpha/2}\right)^2} \tag{2}$$

where

$N$ is the total size of the population;

$n$ is the number of individuals in the sample (minimum sample size);

$Z_{\alpha/2}$ is the critical value that corresponds to the desired degree of confidence;

$p$ is the expected frequency (when this value is not known, 0.50 is adopted);

$q$ is equal to $1 - p$;

$\sigma$ is the population standard deviation of the studied variable (in this example, users);

$d$ is the acceptable error ($d = 5.55\%$, for a significance level of 5%);

For a confidence level equal to 95%, the value of $Z_{\alpha/2}$ is 1.96.

Table 4 shows the minimum sample size (*n*) and the selected sample size (*N*) of questionnaires completed in the FEM environments. The size selected was higher than the minimum size from the statistical point of view.

**Table 4.** The information and questionnaire sample size for different rooms.

| Building | Room | Data Collection Period | Period of the Day | Minimum Sample Size (*n*) | Selected Sample Size (*N*) |
|---|---|---|---|---|---|
| **FEM** | | | Classrooms | | |
| | CL1 | 13, 14 Nov. 2018 | morning | 32.4 | 36 |
| | | | afternoon | 15.3 | 16 |
| | CL2 | 13, 14 Nov. 2018 | morning | 24.9 | 27 |
| | | | afternoon | 17.1 | 18 |
| | | | Offices | | |
| | Of. 3, 4 | 13, 14 Nov. 2018 | morning | 19.7 | 21 |
| | | | afternoon | 12.5 | 13 |
| | Of. 6, 7 | 13, 14 Nov. 2018 | morning | 9.7 | 10 |
| | | | afternoon | 13.4 | 14 |
| | Of. 1, 5, 2 | 13, 14 Dec. 2018 | morning | 9.7 | 10 |
| | | | afternoon | 13.4 | 14 |

## 4. Results

### 4.1. RTQ-C Energy Efficiency Labelling

All blocks achieved level C of energy efficiency in terms of the envelope, except the two transition areas, with level D. This was mainly due to non-compliance with the prerequisites for thermal transmittance of the roofs. The absorptance prerequisite still needs to be met. A possible suggestion to increase the efficiency of the envelope is to paint the roof a lighter colour to meet the prerequisite and increase the energy rating. The details of the label classification level of the buildings are shown in Table 5.

**Table 5.** Classification of the energy efficiency level.

| Block | H | I | J | K | HI | IJ |
|---|---|---|---|---|---|---|
| **Envelope** | C | C | C | C | D | D |
| **Lighting** | B | B | B | C | B | B |
| **Air conditioning** | C | D | E | D | E | E |
| **Score** | 2.68 | 3.87 | 3.03 | 3.44 | 2.22 | 2.38 |
| **General level** | C | B | C | C | D | D |

Regarding the lighting system, most blocks reached level B, mainly due to non-compliance with the prerequisite of automatic lighting and circuit division. Block K was the only one that received a level C for lighting. It should be noted that common fluorescent lamps have been replaced by new LED technologies. Another highlight is that block K received level C for two reasons: first, the total installed power was less than the power limit for the level. Second, at the time of data collection, light bulbs were changed in all blocks except block K. All lamps were replaced by LEDs. It is essential to use lamps with good performance, as there is a difference in the label obtained.

Most received level A in a preliminary step on the label after calculating the installed power in the buildings compared with the power limits. However, as the buildings do not have an automatic shutdown of the circuits, the maximum level that can be obtained is B.

A suggestion to increase efficiency is to install automatic shutdown devices, such as "dimmers" (presence sensors).

The two transition areas and block J received the worst rating, level E, after considering the low performance of the air-conditioning system. Blocks I and K reached level D, and block H reached level C. The old and outdated equipment contributed to low efficiency in some blocks.

Finally, regarding the general level of energy classification, blocks H, J, and K received level C; block I reached level B; and the transition areas obtained level D. It appears necessary, for example, to include some bonuses suggested by the RTQ-C, such as installation of photovoltaic panels, reuse of water, aerators in taps, and flushes with a double activation, to improve the classification levels. In addition, implementing occupancy sensors and painting the roofs with more reflective paint colours could also contribute to greater energy efficiency.

### 4.2. Thermal Comfort

The environmental climate conditions are shown in Figure 6, which describes the inside and outside temperatures where the questionnaires were completed, as well as the relative humidity.

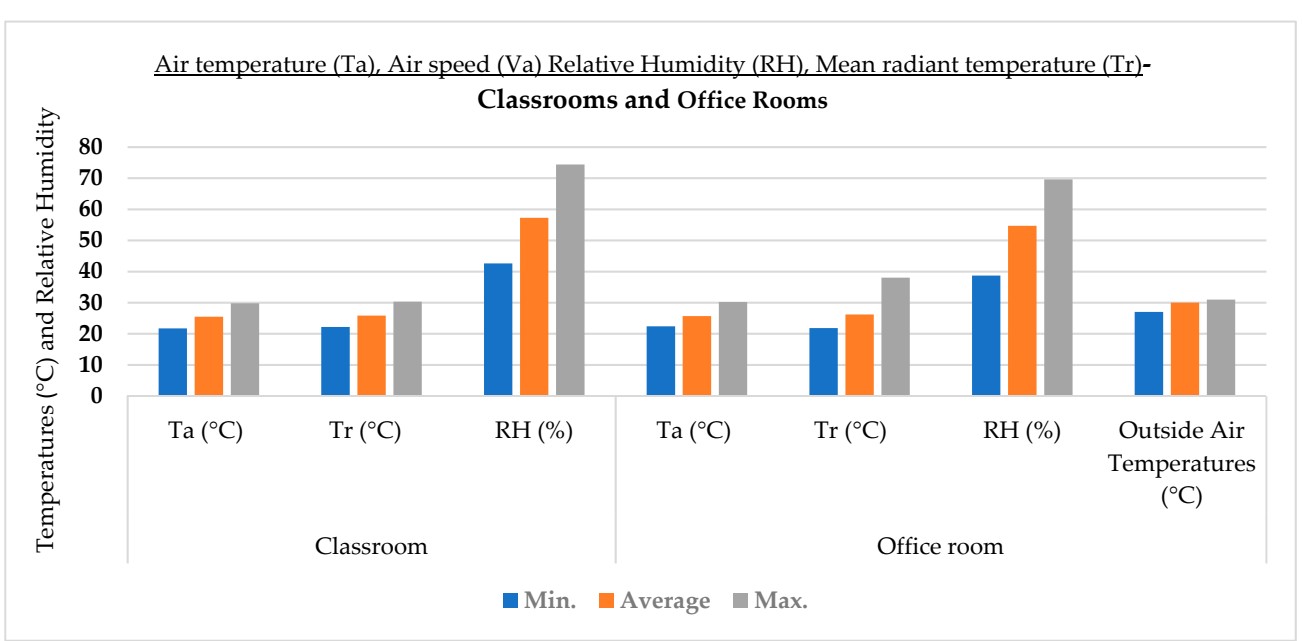

**Figure 6.** The inside environmental data and outside air temperature.

Simultaneously with measuring environmental data, the questionnaires were distributed to users of the environments. A total of 199 questionnaires were completed inside the buildings, 97 in classrooms and 102 in offices.

There were four sessions of questionnaires completed over two days, held at 9:00 a.m., 11:45 a.m., 2:00 p.m., and 4:30 p.m.

The questionnaires covered two classrooms, one on the intermediate floor and another on the upper floor. Most of the selected offices were on the first floor. Some environmental locations are shown in Figure 7a (classrooms) and Figure 7b (office rooms).

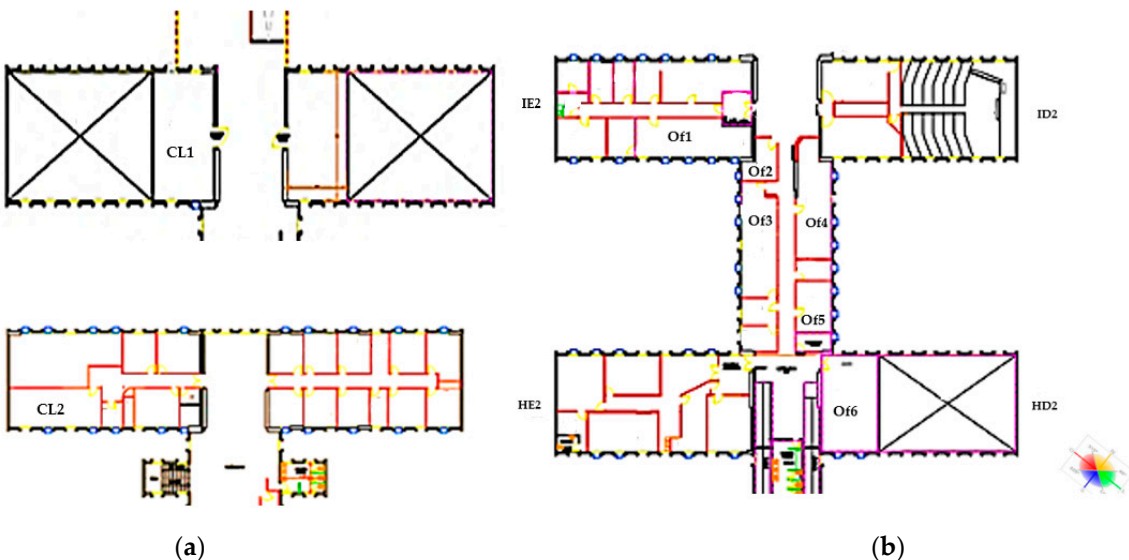

**(a)**                                                                                          **(b)**

**Figure 7.** Location of surveyed buildings on (**a**) classroom and (**b**) office rooms.

Monitoring and data collection occurred during a warm period in Campinas (13 November–14 December) in 2018. Photographs of some environments are shown in Figure 8a–d.

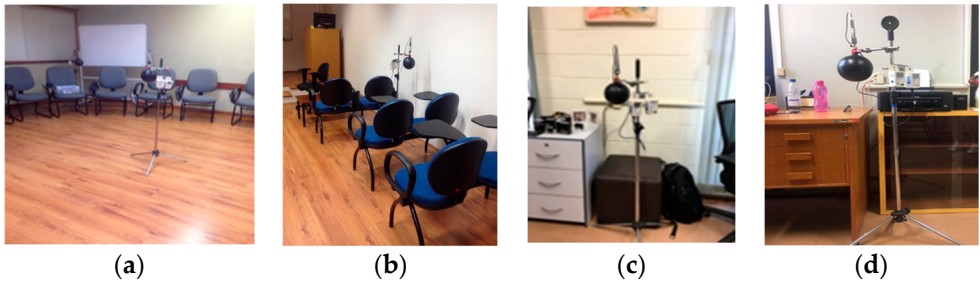

**(a)**                          **(b)**                          **(c)**                          **(d)**

**Figure 8.** (**a**) Classrooms—CL1 and (**b**) CL2. (**c**,**d**) Samples of two offices. (Personal collection.)

The images of two classrooms are shown in Figure 8a,b. CL1 had two façades of windows, but users removed one to fix a blackboard. Thus, the ventilation was not the same as the original layout. Another is CL2, located on the last floor. Figure 8c,d feature office rooms.

Testo model instruments were used for the measurements shown in Figure 9a–c images.

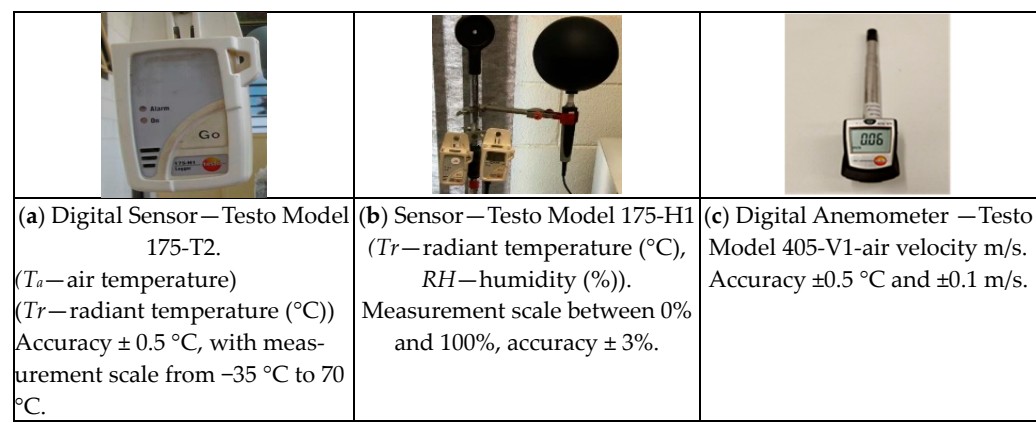

| (**a**) Digital Sensor—Testo Model 175-T2. ($T_a$—air temperature) ($Tr$—radiant temperature (°C)) Accuracy ± 0.5 °C, with measurement scale from −35 °C to 70 °C. | (**b**) Sensor—Testo Model 175-H1 ($Tr$—radiant temperature (°C), $RH$—humidity (%)). Measurement scale between 0% and 100%, accuracy ± 3%. | (**c**) Digital Anemometer —Testo Model 405-V1-air velocity m/s. Accuracy ±0.5 °C and ±0.1 m/s. |
|---|---|---|

**Figure 9.** (**a**) Globe temperature, (**b**) air temperature, (**c**) anemometer.

Photographs of the measurement instruments used in this work and their accuracy ranges are shown in Figure 9.

4.2.1. Classroom Survey Results

Researchers advised users to maintain the same settings in the rooms during the collection phase. Therefore, all data were collected with the air conditioning always on.

The details of the questionnaire numbers in the two classroom environments investigated in the university are shown in Table 6. In CL1, 52 users responded to the questionnaires, and in CL2, there were 45 users. A total of 97 people were interviewed. The percentage of women using the classrooms was 34%, and 66% were men.

**Table 6.** Classroom information.

| Classroom | Total Questionnaires Answered | Percentage (%) | Average Age (Years) |
|---|---|---|---|
| Questionnaires Answered | 97 | 100 | --- |
| CL1 | 52 | 53% | --- |
| Cl2 | 45 | 46% | --- |
| Women | 33 | 34% | 29.4 |
| Men | 64 | 66% | 35.5 |

Figure 10 shows the relationship between the sensation and thermal preferences of users in classrooms and the age groups of users. The thermal sensation of the users indicated a feeling of cold, while they preferred slightly higher temperatures—that is, a slightly warmer environment. This preference for slightly higher temperatures may be an eventual loophole for energy savings in classrooms since they use air-conditioning equipment at the minimum temperature. According to Parra [58], studies by the Brazilian Association of Refrigeration, Air Conditioning and Ventilation (ABRAVA) proved that raising the air-conditioning temperature by two degrees can reduce energy consumption by 7%.

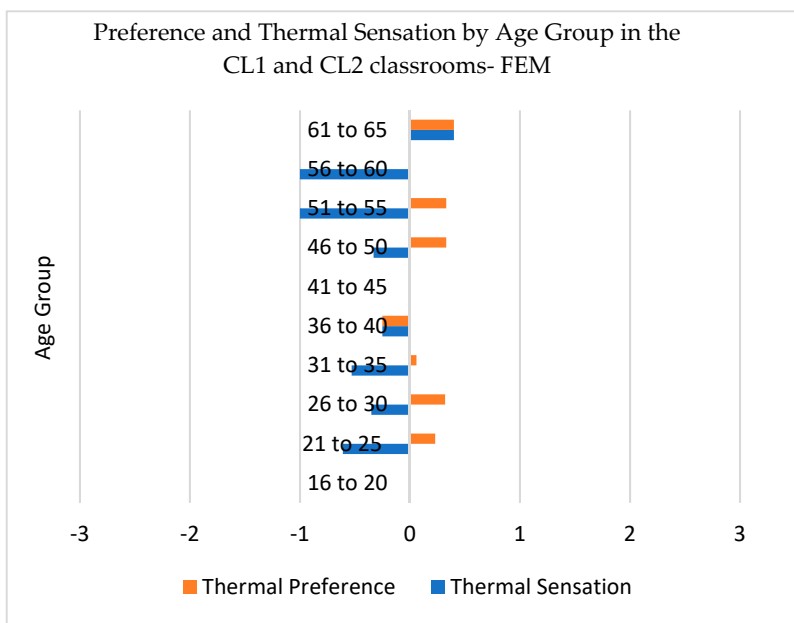

**Figure 10.** Relationship between the thermal sensation and thermal preference of users in classrooms by age group.

The results show an asymmetrical variation in the seven-point scale in classroom environments, indicating heat. However, this indication of heat is due to the room's ambience on the third and top floors. This environment had a more significant impact on the thermal load on the roof of the building, according to the survey, set 2, as seen in Table 2. It had a thermal transmittance on the roof equal to $U^1 = 2.06$ W/m$^2$K | $\alpha^2 = 0.52$. As quoted by Holanda [51], parameters of design (orientation, façades, absorbance, transmittance, etc.) combined with HVAC (air conditioning) can reduce energy consumption by 34%. Users can spend more energy on environmental comfort when some bioclimatic attributes and material performance are not considered during the design and construction phase of a building.

### 4.2.2. PMV/PPD of the Classrooms

Figure 11 shows the results of the PMV-PPD including the two rooms (CL1 and CL2). The results show that the PMV ranged from −1.06, with 28% dissatisfied, to 1.33, with 41% dissatisfied in CL1, which means the results were asymmetric, with a concentration on the positive side that directed to heat. The conditions of Figure 11 occurred with the air conditioning on and according to the usual daily use of the environment.

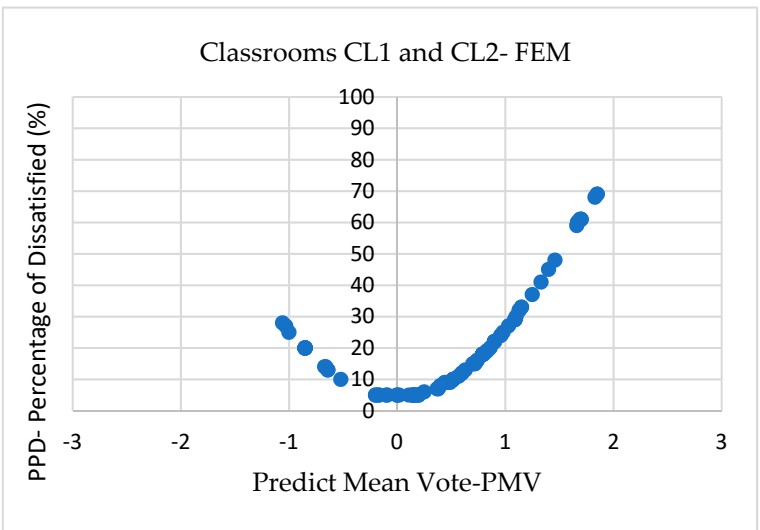

**Figure 11.** Relationship between the PMV and PPD in classrooms.

It is interesting to note that when analysing the same data related to the vote on sensation and thermal preference according to the age group of users, this level of dissatisfaction due to the heat did not appear, replaced instead with a slight dissatisfaction directed at the cold in the classrooms, as shown in Figure 10.

Accordingly, Faraji et al. [38] found that it is also useful to consider aspects such as age and other important parameters that are not considered in some models of comfort analysis when evaluating thermal comfort. In this case, there was a contradiction when evaluating the data calculated with the Comfort 2.03 program and the real average data of sensation and thermal preference by user age. The users voted for slightly cooler but preferred slightly warmer environments, as seen in Figure 10.

However, a separate assessment is also helpful since the rooms' thermal characteristics and locations are distinct.

The result showed the influence of the different locations of the environments on the PMV-PPD on the upper floor.

For CL1, located on an intermediate floor according to the PMV and PPD, there was a PMV value concentration between −1 and +1. However, a tendency towards discomfort due to heat occurred at a rate of approximately 30%, as shown in Figure 12a.

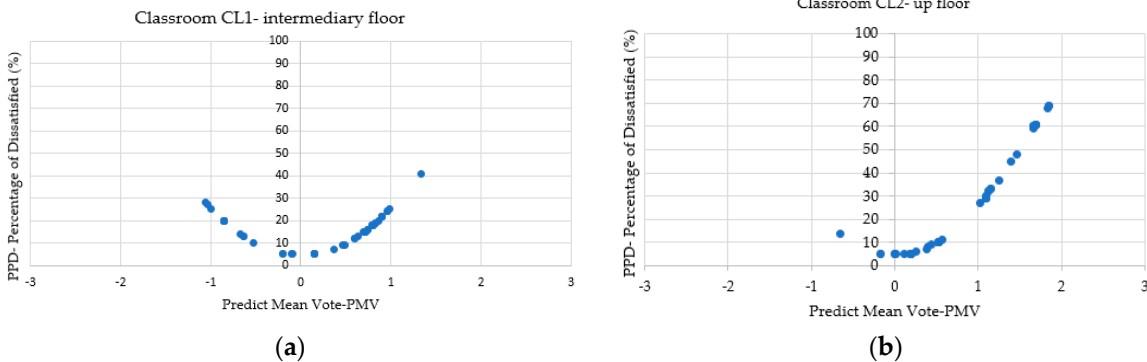

**Figure 12.** Relationship between the PMV and PPD in (**a**) CL1 and (**b**) CL2.

In CL2, Figure 12b, the PMV value concentration was between −0.66 and 1.85. There was a high level of discomfort due to heat, with around 70% of users confirming the expected result of more significant discomfort in proximity to the roof and raising the level of dissatisfaction with the environment. As noted, returning to activities without the previous circulation of internal air to remove the hot air and without mechanical air conditioning makes the environment very uncomfortable.

Tables 7 and 8 show the results for the mean and standard deviation of the calculated variables (PMV and PPD).

**Table 7.** Statistical parameters of the sample data for CL1 (52 questionnaires).

| CL1 | PMV | PPD | (Icl-Clo) |
|---|---|---|---|
| Average | 0.22 | 16.10 | 0.49 |
| Standard Deviation | 0.71 | 7.42 | 0.13 |
| Coefficient of Variation (%) | 322% | 46% | 26% |

**Table 8.** Statistical parameters of the sample data for CL2 (45 questionnaires).

| CL2 | PMV | PPD | (Icl-Clo) |
|---|---|---|---|
| Average | 0.09 | 28 | 0.50 |
| Standard Deviation | 0.80 | 23 | 0.12 |
| Coefficient of Variation (%) | 888% | 82% | 24% |

The coefficient of variation (CV) of the PMV variable was very high in both rooms compared to the other variables.

The coefficient of variation of the PPD variable for CL2, as shown in Table 8, had a much higher value than that for Cl1 in Table 7. Additionally, the average value of the PPD was much higher than that for CL1. Table 7 shows the results, described as the statistical parameters for Cl1 (mean value, standard deviation, and coefficient of variation).

In addition, the values of dissatisfaction perceived by users in CL2 showed more dispersion compared to CL1, which was demonstrated by a comparison of the coefficients of variation between the two rooms.

The comparison between the coefficients of variation (CV) for the PMVs of CL2 and CL1 indicated that the variability in the answers for CL2 was much higher than that for CL1. This can be partly explained by the higher exposure to solar irradiance of JE3 since it is below the top of the building (Figure 2b, block J). In addition to the solar irradiance exposure, the material construction of the roof of building 2 is concrete slab and fibre cement tile with high thermal transfer parameters. CL1 is on the intermediary floor of the building, thus with low solar irradiance exposure, which results in less variability in the perceptions of users.

In the comparison of the averages of the isolation variables (Icl-Clo) of users' clothing in both rooms, the difference in the average was only 0.1 (clo) and 2% with regard to the coefficient of variation between the interviewees of the two environments, according to data shown in Tables 3 and 4.

Table 9 shows the results obtained through the measurements for the two rooms, CL1 and CL2. Based on the data, it was possible to verify that in the first measurement schedule at 9:00, the internal temperature was very close to the external temperature, being very close in both. CL2 was slightly lower. The air-conditioning equipment is usually turned on at the beginning of the workday between 8 and 9 a.m. and turned off during lunch, then turned on again after lunch and turned off again at the end of the workday, only when leaving the room. As shown in Table 9, the temperature in CL2 was higher than in CL1 at almost all times.

**Table 9.** Average internal temperatures of CL1 and CL2 and external average temperature of the building at the evaluated times.

| Data | 9:00 a.m. | 11:30 a.m. | 2:00 p.m. | 4:30 p.m. |
|---|---|---|---|---|
| TBS inside—CL1 | 26.2 °C | 21.7 °C | 24.1 °C | 26.2 °C |
| $T_{ar}$ outside | 27.7 °C | 30.65 °C | 31.5 °C | 32 °C |
| TBS inside—CL2 | 28 °C | 25.7 °C | 30.2 °C | 23.5 °C |

Air conditioners are turned off at 11:30 a.m. and switched on when users return at 2:00 p.m. At that time, the temperature of CL2 again equalled the outside temperature. This situation might affect the roof since, in CL1, there were no significant increases in temperature in the same range.

In CL2, it was possible to notice that the internal temperature at 2 p.m. was close to the external temperature, with a difference of 1.4 °C, which may explain the dissatisfaction with the environment, since the air conditioners were turned off during the lunch break activities, restarting again at 2 p.m. in a scorching environment.

4.2.3. Office Rooms

The offices were evaluated according to actual use; some environments were designed for individual use, while others were shared. The evaluation was also performed according to daily use and oriented to maintain use. Therefore, all data were collected with the air conditioning always on.

Questionnaires

In the office room environments, 102 users responded to the questionnaire. The percentage of women using the office room environment was 62%, and for the men, it was 38% (Table 10). The data details of the office room environments investigated in the university are shown in Table 10.

**Table 10.** Data collected with the questionnaires in offices.

| Office Rooms | Total | Percentage (%) | Average Age (Years) |
|---|---|---|---|
| Questionnaires returned | 102 | 100 | 47 |
| Woman | 63 | 62% | 45.7 |
| Man | 39 | 38% | 49.2 |

The average age of all office users was 47 years, with women's average age 47.5 years and men's 49.2 years.

Figure 13 shows the relationship between thermal sensation and thermal preference by age group of users in the classrooms. The lowest age found for users in the FEM office environments was 24 years old, and the maximum was 65 years old. In these office environments, there were no users in the following three age groups: 16 to 20, 26 to 30, and 36 to 40 years old (Figure 13).

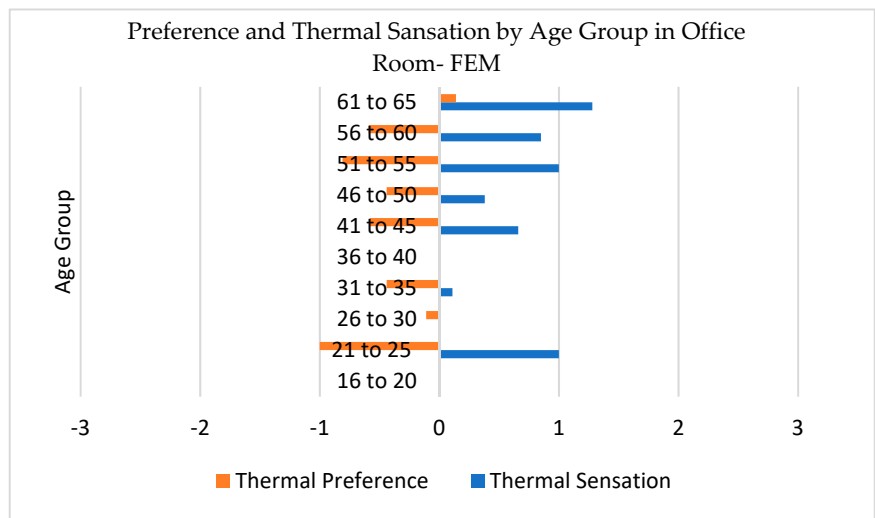

**Figure 13.** Relationship between the thermal sensation and thermal preference of users in office rooms by age group.

The age groups with the highest representation in the offices were, in ascending order, the 31- to 35-year age group and the 46- to 50-year age group. The second largest group was between 51 and 55 years old, and the third was between 56 and 60 years. Notably, all users of the surveyed environments participated in the survey. According to the survey, "They insisted on participating because they considered the environment uncomfortable".

The thermal sensation vote ranged from 0.11 in the 31- to 35-year age group, 0.38 in the 46- to 50-year age group, and 1 in the 51- to 55-year age group to 1.28 in the 61- to 65-year age group.

The average thermal preference vote in these same age groups was −0.44 in the 31- to 35-year and 46- to 50-year age groups. The average thermal preference vote in the 51- to 55-year age group was −0.81. In the 56- to 60-year age group, it was −0.59.

Environments remained wholly closed during the night. When arriving early, users turn on the air-conditioning equipment and leave it on until lunchtime at noon. The equipment is turned off during the lunch interval. The devices are turned on again when users return to their work environment and are only turned off at the end of the day. Additionally, as observed in the study, a relatively common case at the university is opening the windows and "forgetting", leaving them open for a long time or even all day, with the air conditioning on.

### 4.2.4. PMV/PPD—Office Rooms

The predicted mean value of the thermal sensation votes of the respondents inside the FEM office rooms ranged from neutral (−0.16) to very hot (+2.61), which is asymmetric, indicating a warmer environment.

In Figure 14, asymmetry in the PMV scale ranging from −0.16 to +2.91 in office environments is shown. Such asymmetry indicates dissatisfaction due to heat on the part of users. There was a higher concentration between +0.13 and +1.32. The PPD varied between −0.16 at 5% and 2.91 at 95%, concentrated between 5% and 41%.

**Figure 14.** The PPD and PMV in the office rooms of the FEM.

When evaluating the Of1 and Of6 environments in the NW orientation, the PMV range was −0.15 to 1.03, and the PPD was between a minimum of 5% and a maximum of 95%. In Of2 and Of3, located in the NW orientation, the PMV range was between −0.15 and 1.03, and PPD had a minimum of 5% and a maximum of 27%. As for Of4 and Of5, located in the NE direction, the PMV range was between 0.02 and 1.03, with the PPD a minimum of 5% and a maximum of 27%. As can be seen, the level of dissatisfaction was higher in the SE direction. In Campinas, the predominant winds are in the (SE) direction. The discomfort in these rooms can be due to one of three reasons:

(a) The lack of ventilation in the environment is repeatedly claimed by users of this type of campus building, which currently comprises approximately 80 buildings;

(b) In the specific buildings studied, there is a sequence of conjugated blocks that can worsen the lack of ventilation and increase discomfort;

(c) The poor quality and lack of performance of the air-conditioning equipment are due to ageing degradation, dating back to the 1980s. The statistical parameters PMV, PPD, and Iclo-clo of the occupants of office rooms are shown in Table 11. When compared to classrooms CL1 and CL2, the following differences and similarities can be highlighted:

(a) The PMV of offices (Table 11) was higher, more than double the PMV of CL1 and CL2;

(b) The variability in the PMV (measured by the coefficient of variation: C.V.) for the offices (C.V. = 94.23%) was much lower than those for CL1 (C.V. = 322%) and CL2 (C.V. = 888%);

(c) The PPDs (predicted percentages of dissatisfied) for the offices were very similar to those for CL1, although significantly lower than for CL2;

(d) The Iclo-clo values of all rooms were very similar, around 0.5.

**Table 11.** Statistical parameters of the sample data for offices (102 questionnaires).

| Office | PMV | PPD (%) | Iclo-clo |
|---|---|---|---|
| Average | 0.52 | 14.59 | 0.48 |
| Standard Deviation | 0.49 | 15.28 | 0.12 |
| Coefficient of Variation (%) | 94.23 | 104.73 | 25 |

Possible explanations for these findings highlighted above from the data in Table 11 are as follows:

(i) The air conditioning in the offices is older, with worse performance than that in CL1 and CL2. In addition, the worse behaviour of office occupants related to opening and closing the windows explains the results described in item (a) above;

(ii)   The gender representation in CL1 and CL2 contains fewer women (34%) and an average age of 29.4 years, while, in offices, women are the majority (62%), with an average age of 45.7 years. These significant differences (gender and age) among the groups of occupants are a possible source of the variability in the PMV in offices, as highlighted in item (b) above;

(iii)  As mentioned before, CL2 has higher exposure to solar irradiance since it is below the roof of the building (Figure 2b, block J). Furthermore, the material construction of the roof of building 2 is concrete slab and fibre cement tile with high thermal transfer parameters. This explains the higher PPD (predicted percentage of dissatisfied) for CL2 compared to the offices and CL1 (intermediate floor), which was highlighted in item (c) above;

(iv)  The effect of clothing did not influence thermal comfort in these selected indoor spaces, since for all rooms (CL1, CL2, and offices), the occupants' daily habits were well adapted to climate conditions during warm periods. They are familiar with the proper clothing for these spaces. This can explain the very similar Icl-clo values for all rooms, as shown in Table 11 and highlighted in item (d) above.

In the period evaluated, the outside air temperature average fluctuated during the day relative to the time evaluated; nonetheless, the inside air temperature with the air conditioning on was nearly the same all day, as shown in Table 12. The highest temperature of 26 °C was experienced at 4:30 p.m.

**Table 12.** Average temperatures inside and outside the building.

| Ta | 9:00 a.m. | 11:30 a.m. | 2:00 p.m. | 4:30 p.m. |
|---|---|---|---|---|
| Average air temperature (Ta) outside | 27.7 | 30.65 | 31.5 | 30.9 |
| Average Ta inside | 25.81 | 25 | 25.75 | 26 |

The findings of our research work have some similarities and, of course, differences compared to other studies investigating thermal comfort based on surveys in university buildings. We selected seven studies with university classroom surveys for comparison, including countries with distinct climates and student behaviours, such as Australia (Alghamdi et al. [58]), China (Jing et al. [59]; Zhang et al. [7]), Italy (Nico et al. [60]), Ecuador (Guevara et al. [40]), Indonesia (Hamzah et al. [61]), and Brazil (Niza et al. [62]). One selected topic was related to comparisons between the PMV and PPD votes collected in classes with the votes predicted by models. We found significant differences in our research work, although no systematic trend was found towards the overestimation or underestimation of predicted votes by the model. Hamazah [61] and Guevara [40] also found significant differences, both employing questionnaires in warm regions (Indonesia and Ecuador, respectively). They mentioned that people in tropical regions are more tolerant of higher temperatures. Also, significant differences were reported by Jing et al. [59] in a Chinese region in a cold climate zone. Nevertheless, Nico et al. [60] and Alghamdi et al. [58] found very similar results between model predictions and collected data. Another selected topic was related to gender differences in thermal sensation. Most authors reported differences, although slight, with women's preferences towards warmer environments. These slight differences were also found in our research at UNICAMP, Brazil.

Most authors recognise the need for further studies based on surveys and questionnaires applied to university students, in both air-conditioned and free-running environment buildings, to better understand students' thermal perceptions in different classrooms. These initiatives can improve the design and performance of university buildings and enhance student well-being and productivity.

## 5. Conclusions

This research project conducted a thermal comfort analysis that included on-site measurements of indoor thermal conditions and a survey of the building users. The

PMV-PPD evaluation comparison was applied to occupants of two classrooms and seven office rooms inside university buildings and allowed an explanation of the influence of construction and architecture parameters (i.e., the position of the rooms inside the building, exposure to solar irradiance, wall and ceiling materials, etc.), air-conditioning equipment, and occupant behaviour on the thermal comfort of the room users. These parameters impacted not only the average values (i.e., PMV, PPD) but also the variability of such values among the users.

The comparison of the survey and calculation results among the selected rooms also revealed the influence of gender and age on thermal comfort perception, with impacts both on the average values of PMV-PPD and on the variability of the survey results.

Furthermore, room users showed significant levels of dissatisfaction with the environment, around 95% due to heat, even with the air conditioning on in the offices. In this sense, some aspects were found that may be related to the low thermal performance of the buildings, as follows:

(i) The inappropriate capacity of the air-conditioning equipment related to the environment;
(ii) The inappropriate use of the air-conditioning equipment;
(iii) The inappropriate use of construction technologies (opening and closing windows and doors);
(iv) The need to improve and simplify user ability to change the equipment setpoint;
(v) Allowing the change of the worktable position inside the offices to achieve better performance and thermal comfort for users;
(vi) The location, types, and quality of materials.

All of these elements can enhance building performance for the users. On the other hand, users should pay more attention to turning on the air conditioner and closing the windows and doors at the right moments.

Given the collected data, survey results, and discussions, it is clear that interventions in buildings are necessary to mitigate the low energy efficiency achieved in the selected buildings of the University of Campinas. Due to the architectural orientation and the use of spaces, interventions should focus on the internal environments to the exclusion of the external ones since the area around the buildings already has vegetation with no prospect of expansion.

It is important to highlight that the sustainable campus project partially described in this article contributed to several initiatives undertaken by the University of Campinas, including (a) a progressive integration of energy efficiency standards into the existing building management and design of new buildings; (b) large-scale programme replacement of old air-conditioning equipment and lighting systems inside and outside the buildings; and (c) campaigns encouraging awareness and behaviour changes in university building occupants.

**Author Contributions:** Conceptualisation, E.F.T.B. and L.C.L.; methodology, E.F.T.B., L.C.L. and A.P.A.S.C.; formal analysis, E.F.T.B., A.P.A.S.C. and F.S.D.L.; investigation, E.F.T.B. and A.P.A.S.C.; writing—original draft preparation, E.F.T.B. and A.P.A.S.C.; supervision, L.C.L. All authors have read and agreed to the published version of the manuscript.

**Funding:** This research was funded by the CPFL Energy Company-Companhia Paulista de Força e Luz, grant number 00063-3032/2017-PA3032.

**Institutional Review Board Statement:** Not applicable.

**Informed Consent Statement:** This study was conducted in accordance with Ethical Conduct in Human Research from the Research Ethics Committee (CEP) of the University of Campinas-UNICAMP (2018) and approved by the Ethics Committee of the University of Campinas, São Paulo, CAAE: 87220518.3.0000.8142 and authorisation (protocol code 3.087.904 and date of approval: 17 December 2018; home page: "http://www.prp.unicamp.br/index.php/comite-de-etica-em-pesquisa/ (accessed on 10 October 2023)".

**Data Availability Statement:** The datasets presented in this study are available on request to the corresponding author, subject to the approval of the CPFL Brazil.

**Acknowledgments:** This work was developed under the Electricity Sector Research and Development Program "Sustainable campus model at the University of Campinas-Brazil: An integrated living lab for renewable energy, electric mobility, energy efficiency, monitoring and energy demand management", regulated by the National Electricity Agency (ANEEL in Portuguese), in partnership with CPFL Brazil (Local Electricity Distributor). Acknowledgement is due to the School of Mechanical Engineering for supporting the research with field study.

**Conflicts of Interest:** The authors declare no conflict of interest.

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
