# Peer review of "Energy Efficiency and Thermal Comfort Analysis in a Higher Education Building in Brazil"

_sustainability, doi:10.3390/su16010462_

Round 1

Reviewer 1 Report

Comments and Suggestions for Authors

-       The English needs editing.

-       P 7: Please add the reference to Table 1.

-       Section 3.4.1: Usually measuring devices are put at 60 cm height for sitting person. Please indicate the standard according to which the devices were placed at 1.1 m for sitting person.

-       There are two “Figure 10 & 11” in the text.

-       Were the windows open while the questionnaires were filled out?

-       Seven office rooms with various orientations are studied, but the difference between results obtained from these offices is not discussed clearly.

Comments on the Quality of English Language

 The English needs editing.

Author Response

See comments and answers in the attached file. 

Reviewer 2 Report

Comments and Suggestions for Authors

Dear Authors,

as the publication has already been reviewed with me the first time, I will try to refer to those previous comments. I have checked that the authors have taken all my previous comments into account and have corrected their work. Thank you for this.

As a new remark, please prepare a discussion paragraph, and compare your results and your publication to other similar works on the subject. It is important that such a paragraph appears in the paper, before the conclusion paragraph.

Author Response

(The authors gave the same response as above.)

Reviewer 3 Report

Comments and Suggestions for Authors

Please double check the existing grammatical issues in the paper.

Author Response

(The authors gave the same response as above.)
